# Juvenile Heat Tolerance in Wheat for Attaining Higher Grain Yield by Shifting to Early Sowing in October in South Asia

**DOI:** 10.3390/genes12111808

**Published:** 2021-11-18

**Authors:** Uttam Kumar, Ravi Prakash Singh, Susanne Dreisigacker, Marion S. Röder, Jose Crossa, Julio Huerta-Espino, Suchismita Mondal, Leonardo Crespo-Herrera, Gyanendra Pratap Singh, Chandra Nath Mishra, Gurvinder Singh Mavi, Virinder Singh Sohu, Sakuru Venkata Sai Prasad, Rudra Naik, Satish Chandra Misra, Arun Kumar Joshi

**Affiliations:** 1Borlaug Institute for South Asia (BISA), NASC Complex, DPS Marg, New Delhi 110012, India; u.kumar@cgiar.org; 2International Maize and Wheat Improvement Center (CIMMYT), NASC Complex, DPS Marg, New Delhi 110012, India; 3International Maize and Wheat Improvement Center (CIMMYT), El Batan 56237, Mexico; R.Singh@cgiar.org (R.P.S.); s.dreisigacker@cgiar.org (S.D.); J.CROSSA@cgiar.org (J.C.); s.mondal@cgiar.org (S.M.); l.crespo@cgiar.org (L.C.-H.); 4Leibniz Institute of Plant Genetics and Crop Plant Research (IPK), 06466 Gatersleben, Germany; roder@ipk-gatersleben.de; 5Campo Experimental Valle de Mexico-INIFAP, Carretera los Reyes-Texcoco, Coatlinchan 56250, Mexico; j.huerta@cgiar.org; 6ICAR-Indian Institute of Wheat and Barley Research (IIWBR), ICAR, Karnal 132001, India; gyanendrapsingh@hotmail.com (G.P.S.); Chandra.Mishra@icar.gov.in (C.N.M.); 7Plant Breeding and Genetics Department, Punjab Agricultural University, Ludhiana 141004, India; mavig666@pau.edu (G.S.M.); sohuvs@pau.edu (V.S.S.); 8Regional Research Station, Indian Agricultural Research Institute, Indore 542001, India; sprasad98@gmail.com; 9Department of Genetics and Plant Breeding, University of Agricultural Sciences, Krishi Nagar, Dharwad 580005, India; rvnaikgpb@gmail.com; 10Genetics and Plant Breeding Group, Agharkar Research Institute, Pune 411004, India; wheat.bisa@gmail.com

**Keywords:** early heat stress, *Triticum aestivum*, heat tolerance, VRN, PPD, photoperiod

## Abstract

Farmers in northwestern and central India have been exploring to sow their wheat much earlier (October) than normal (November) to sustain productivity by escaping terminal heat stress and to utilize the available soil moisture after the harvesting of rice crop. However, current popular varieties are poorly adapted to early sowing due to the exposure of juvenile plants to the warmer temperatures in the month of October and early November. Therefore, a study was undertaken to identify wheat genotypes suited to October sowing under warmer temperatures in India. A diverse collection of 3322 bread wheat varieties and elite lines was prepared in CIMMYT, Mexico, and planted in the 3rd week of October during the crop season 2012–2013 in six locations (Ludhiana, Karnal, New Delhi, Indore, Pune and Dharwad) spread over northwestern plains zone (NWPZ) and central and Peninsular zone (CZ and PZ; designated as CPZ) of India. Agronomic traits data from the seedling stage to maturity were recorded. Results indicated substantial diversity for yield and yield-associated traits, with some lines showing indications of higher yields under October sowing. Based on agronomic performance and disease resistance, the top 48 lines (and two local checks) were identified and planted in the next crop season (2013–2014) in a replicated trial in all six locations under October sowing (third week). High yielding lines that could tolerate higher temperature in October sowing were identified for both zones; however, performance for grain yield was more promising in the NWPZ. Hence, a new trial of 30 lines was planted only in NWPZ under October sowing. Lines showing significantly superior yield over the best check and the most popular cultivars in the zone were identified. The study suggested that agronomically superior wheat varieties with early heat tolerance can be obtained that can provide yield up to 8 t/ha by planting in the third to fourth week of October.

## 1. Introduction

Wheat is a strategic, staple crop in India [1] and South Asia [2]. It holds particular significance for women from marginal and small farming households, who contribute significantly to wheat production systems and livestock management. The Indo-Gangetic Plains of South Asia are considered crucial for meeting foot security needs of a huge population of >900 million people. The total population of the four South Asian countries (India, Nepal, Bangladesh and Pakistan) is 1971 million as of November 2021 (http://www.worldometers.info/world-population/southern-asia-population/, accessed on 11 November 2021). At present, the Gangetic Plains of South Asia are considered optimal for wheat farming, but may become sub-optimal by 2050 due to climate change [3]. On top of that, if the speed of decline in ground water table remains as it is today, this may be a major cause of food insecurity and affect the livelihood of farmers [4]. Heat tolerance of crops varies greatly, and wheat is among the most sensitive of the major staples. A study reported that yield losses in wheat for each 1 °C temperature may fall between 3 and 17% for north western India and Pakistan [5].

To sustain wheat productivity, farmers in northwestern and central India are shifting to earlier sowings (in the second fortnight of October, immediately after rice) to take benefits of residual moisture of the previous monsoon and to allow crop to mature much earlier than the end of March to beginning of April, a period when terminal heat stress becomes a major issue. In the 10 million ha rice-wheat system of South Asia, there is often adequate residual soil moisture at the end of the irrigated rice season, but these soils dry out by mid-November, the current optimum sowing date, requiring additional irrigation to achieve uniform wheat germination and establishment [6]. Furthermore, water tables cyclically fall so low in central India by late January that pumping is prohibitively expensive or not possible [7]. Early sowing using residual moisture allows farmers to save one irrigation and thus increases water productivity, especially when combined with other agronomic practices. However, most of the present wheat cultivars in farmers fields are not well adapted to early season sowing (3rd week of October), since temperatures are quite warm in this period and affect wheat crop by forcing it to grow much faster, accumulate lesser biomass and produce lower yield. Adapted varieties will need to be tolerant to early as well as late-season warmer temperatures [8].

The present knowledge on genetic control of adaptation response and heat tolerance is lacking. Major genes underlying adaptation have been identified in recent years, namely those controlling vernalization response, photoperiod sensitivity and development rate “earliness per se” [9,10,11,12]. However, the major gaps in the knowledge of the effects of possible allele combination on adaptation in specific environments. The *Vrn* genes control difference of spring and winter wheat by defining the chilling hours required by the wheat plant to be able to flower, while the *Ppd* genes play an important part in delaying flowering time in the spring after vernalization requirement has been satisfied. The *Eps* loci may influence more subtle effects in the life cycle for regional adaptation. QTL studies have described various regions of potential *Eps* genes [13,14] with some chromosomes (4B, 6A, 7D) showing this more frequently in a diverse germplasm [15]. *Vrn*, *Ppd* and *Eps* genes additionally present epistatic interactions [16,17]. Large numbers of allele combinations will therefore be involved in determining the regulation of growth habit and optimal adaptation to a certain environment.

CIMMYT’s contribution to the Green Revolution is well known. Still today, advanced germplasm lines developed and distributed through international trials and nurseries by CIMMYT are grown annually on more than 100 locations in the world [18]. Analyses of some of these International Yield Trials indicate that grain yields of the best new genotypes are significantly higher than the local checks [19,20]. However, no efforts were made so far to breed for early heat tolerance under October sowing. Therefore, the present investigation was initiated to identify wheat varieties and adapted germplasm for early (October) sowing under warmer temperatures that exploit the residual moisture and escape terminal heat stress.

## 2. Materials and Methods

The term ‘early (October) sowing’ used in this manuscript refers to sowing under third week of October. Weather data were recorded for the three years of experimentations in Northwest India is given in Figure 1 as an example.

### 2.1. Experiment 1: Field Trials of 3222 Diverse Wheat Genotypes

A set of 3322 diverse high-yielding bread wheat lines showing a range of maturity and improved wheat varieties were prepared at CIMMYT headquarter in Mexico and provided to national partners. The details of lines with pedigree and other information is provided as supplementary file (Appendix A). The lines were planted in an augmented design in six locations (Ludhiana, Karnal, New Delhi, Indore, Pune and Dharwad) spread over NWPZ and CPZ of India under early (October) sowing in year 2012. The date of sowing in these locations varied from October 19–22. Five checks (Super 152, PBW 343, Baj, Munal#1 and Danphe#1) were planted every 20th plot throughout the trial. Since the objective was to identify wheat lines that can demonstrate superior agronomic performance under early (October) sowing, no trial was conducted under normal (November) sowing.

Planting of trial was done by hand in paired rows/line by keeping plot size of 2 m length. Row to row spacing was 25 cm while plant to plant by 3–5 cm at all the locations. The agronomic practices adopted were those recommended for normal fertility (120 kg N: 60 kg P_2_O_5_: 40 kg K_2_O ha^−1^). As per recommended practice, K_2_O and P_2_O_5_ were applied only at the time of sowing, while nitrogen was split in to three stages: 60 kg N ha^−1^ at sowing, 30 kg N ha^−1^ at first irrigation (21 days after sowing) and 30 kg N ha^−1^ at the second irrigation (45 days after sowing). A total of 5 irrigations were given in NWPZ trials while 4 were given in CPZ, the first being on the 21st day after sowing. Data were recorded for five traits: days to heading (DH), plant height (PH), thousand-grain weight (TGW), grain yield (GY) and canopy temperature (CT).

### 2.2. Experiment 2: Genotyping of Germplasm for Ppd and Vrn Genes

To understand the distribution of *Ppd* and *Vrn* genes in the germplasm evaluated for early (October) sowing, these genotypes were screened for *Ppd* and *Vrn* genes described below. This was achieved for 3209 lines tested in the first year. This included 2748 lines from CIMMYT and 461 lines from national programs.

The DNA samples were provided by CIMMYT, while molecular mapping was done at Leibniz Institute of Plant Genetics and Crop Plant Research (IPK), Germany. 

#### 2.2.1. Genotyping of Ppd-D1

The photoperiodism gene *Ppd-D1* on chromosome 2DS is the major photoperiod response locus in wheat and codes for a gene of the pseudo-response regulator (PRR) family [9]. A semi-dominant mutation widely used in the ‘green revolution’ converts wheat from a long day (LD) to a photoperiod insensitive (day neutral) plant. Varieties with the photoperiod insensitive *Ppd-D1a* allele, which causes early flowering in short day (SD) or LDs had a 2 kb deletion upstream of the coding region [9]. Specific primers monitoring the presence or absence of this deletion can be used to distinguish wildtype plants (photoperiod sensitive) from plants carrying the mutant allele *Ppd-D1a* (photoperiod insensitive). Therefore, genotyping of all the lines was done with the *Ppd-D1* specific primers (Appendix A).

#### 2.2.2. Genotyping of Vrn-1 Genes

The series of Vrn-1 genes, Vrn-A1, Vrn-B1 and Vrn-D1 is located on chromosomes 5AL, 5BL and 5DL, respectively. The gene product is most likely the MADS box gene AP1 [21]. The Vrn genes determine the vernalization requirement for wheat. In winter wheat, all three loci are usually found in recessive state, while in spring wheat one or several loci contain dominant alleles [22]. The INDEL markers specific to all three genes were available, which were used in genotyping of CIMMYT and country lines (Appendix A).

#### 2.2.3. Genotyping of Vrn-B3

The vernalization gene *VRN3* encodes a RAF kinase inhibitor-like protein with high homology to *Arabidopsis* protein *FLOWstERING LOCUS T* (FT) [23,24]. Gene *Vrn-B3* is located on the short arm of wheat chromosome 7B. It determines, besides the loci *Vrn-A1*, *Vrn-B1* and *Vrn-D1*, the spring or winter-type of wheat varieties. In winter wheat, all four genes are usually present in a recessive state [22]. When one or several of the four loci are present in a dominant state, the variety can be considered a spring variety. The genotyping was conducted according to the protocol of [22] (Appendix A).

#### 2.2.4. Genotyping of Photoperiod Insensitive Ppd-A1a Mutations

In hexaploid wheat, mutations conferring photoperiod insensitivity have been mapped on the 2B (*Ppd-B1*) and 2D (*Ppd-D1*) chromosomes. The mutation in A-genome of hexaploid wheat is lacking so far. However, the mutations by deletions in PRR (pseudo response regulator) gene of A-genome in tetraploid wheat were associated with photoperiod insensitivity. We applied a marker set developed for tetraploid wheat [25] to the CIMMYT and country lines (Appendix A).

#### 2.2.5. Genotyping of SSR-Marker GWM4167 Associated to Ppd-B1

Since, for the gene *Ppd-B1*, no gene specific markers were available during genotyping, we decided to test a linked SSR-Marker. In the study by Zanke et al. (2014), it was shown that the SSR-marker allele GWM4167-217bp was associated with a delay in flowering in European winter wheat. GWM4167 maps to a similar position like *Ppd-B1* and it was assumed that the effect may be due to *Ppd-B1* [26].

#### 2.2.6. Genotyping of SSR-Marker GWM291 Associated to Vrn-A2

*VRN2* is a dominant repressor of flowering and down-regulated by vernalization. The *VRN2* region includes two similar ZCCT genes encoding proteins with a putative zinc finger and a CCT domain that have no clear homologs in Arabidopsis [23,24]. *Vrn-A2* is located on the distal end of chromosome 5AL, and no mutants derived from this gene have been described as markers. However, a strong effect of the microsatellite allele GWM291-176bp for decreasing the time to heading was described in European winter wheat [26]. Since GWM291 is in the same chromosomal region as *Vrn-A2*, the effects were attributed to *Vrn-A2* [26].

#### 2.2.7. Marker Ppd-B1_R36-F31 Detecting Copy Number Variation at Locus Ppd-B1

For the insensitivity locus *Ppd-B1* on chromosome 2BS so far, no candidate mutations in the gene sequence have been described. Recent research showed that alleles with an increased copy number of *Ppd-B1* confer an early flowering day neutral phenotype [27]. Specific PCR primers detecting the junction between intact *Ppd-B1* copies as described in the varieties ‘Sonora’ and ‘Timstein’ were used to identify varieties with several copies of *Ppd-B1*.

### 2.3. Experiment 3: Field Trials of Adapted Breeding Materials under October Sowing in NWPZ and CPZ of India in Crop Season 2013–2014

Wheat lines, including local check, found adapted and high-yielding for early sowing in Northwestern Mexico, and NWPZ and CPZ of India were tested in year 2013–2014 in replicated yield trials at the same six locations in India to identify best adapted germplasm. Each trial comprised of 48 genotypes and two local checks that were the best locally adapted varieties in the two zones. Each trial had 3 replications and was arranged in an α lattice design. Planting was done on the second fortnight of October with date of sowing falling between October 19–22. Standard plot size of 6 rows of 6 m with row to row spacing of 20 cm was used. Agronomic management was used as described in experiment 1. Data was recorded for grain yield (GY) and for other traits; days to heading (DH), days to maturity (DM), plant height (PH) and thousand grains weight (TGW). Like experiment 1, the objective was to identify wheat lines that can demonstrate superior agronomic performance under early (October) sowing, hence no trial was conducted under normal (November) sowing.

### 2.4. Experiment 4: Field Trials of Adapted Breeding Materials under October Sowing in NWPZ of India in Crop Season 2014–2015

Since performance of wheat lines was much better in NWPZ under October sowing, a set of 30 lines was tested in crop season 2014–2015 in replicated yield trials at the same three locations of NWPZ India (Karnal, Ludhiana and Hisar) to further identify adapted genotypes for October sowing. Each trial was comprised of 28 genotypes and two local checks (HD 2967 and DPW 621-50) that were the best locally adapted varieties in NWPZ. Each trial had 3 replications and was arranged in an α lattice design. Planting was done on the second fortnight of October, i.e., between October 19–21. Standard plot size of 6 rows of 6 m with row to row spacing of 20 cm was used. Agronomic management was used as described in the previous experiment. Observations were recorded for yield and yield traits as mentioned in experiment 3.

### 2.5. Statistical Analysis

For each trial, data analysis was done using R software following a mixed model approach and using the adjusted means for each genotype at individual as well multiple locations in each time of the year. In the analysis of variance and the variance estimates, the model was as shown below.
Y = checks + location + checks × location + genotypes + genotypes × location + error

Least square means were estimated for the lines in the trial. The mean grain yield of a genotype was also expressed as percent of the local check using the following formula:$$\%GY=\left(\frac{GYg}{GYc}\right)\times 100$$
where *GYg* is the mean grain yield of a line and *GYc* is the mean grain yield of the local check.

The analysis of variance for all traits was done together in the first year using all six locations. However, in the next two years, analysis was done separately for each zone for each of the two years using data of three locations for each of the two zones. To identify superior lines across locations, we performed stability analysis using the SREG model [28] for the response of the lines on the combination of the 5 sites for yield, and yield related traits (DH, DM, PM and TGW).

The sites regression model (SREG) was:(1)$${\bar{y}}_{ij.}=\mu_{j}+\sum_{k=1}^{t}\lambda_{k}\alpha_{ik}\gamma_{jk}+{\bar{\varepsilon }}_{ij.}$$
where, ${\bar{y}}_{ij.}$ is the mean of the ith cultivar in the jth environment for g cultivars and e sites (i = 1, 2, …, g and j = 1, 2, …, e); $\mu_{j}$ is the site mean; $\lambda_{k}$ ($\lambda_{1}\geq \lambda_{2}\geq \ldots \geq \lambda_{t}$) are scaling constants (singular values) that allow the imposition of orthonormality constraints on the singular vectors for cultivars, $\alpha_{k}$ = ($\alpha_{1k},\ldots ,\alpha_{gk}$)′ and sites, $\gamma_{k}$ = ($\gamma_{1k},\ldots ,\gamma_{ek}$)′, such that $\sum_{i}\alpha_{ik}^{2}=\sum_{j}\gamma_{jk}^{2}=1$ and $\sum_{i}\alpha_{ik}^{}\alpha_{ik’}^{}=\sum_{j}\gamma_{jk}^{}\gamma_{jk’}^{}=0$ for k ≠ k′; $\alpha_{ik}^{}\;and\;\gamma_{jk}^{}$, for k = 1, 2, 3, …, are called “primary,” ”secondary,” “tertiary,” …, effects of ith cultivar and the jth site, respectively; ${\bar{\varepsilon }}_{ij.}$ is the residual error assumed to be normally and independent distributed with 0 means and variance $\sigma^{2}/r$ (where $\sigma^{2}$ is the pooled error variance and r is the number of replicates). The number of bilinear terms is t ≤ min (g, e). Estimates of the multiplicative parameters in the kth bilinear term are obtained as the kth component of the deviations from the additive part of the model. In the SREG model, only the main effects of cultivars plus the G×E are absorbed into the bilinear terms.

## 3. Results

### 3.1. Performance of 3322 Wheat Genotypes Set for Agronomic Traits in the Early (October) SOWN Conditions

The 3322 genotypes that included CIMMYT breeding lines and spring wheat cultivars collected from different countries, displayed significant variation for grain yield and yield traits: DH, DM, PH and TGW (Table 1). The average range of different traits over locations was very high, for instance 1.1–8.2 t/ha for grain yield, 53–112 for DH, 70–143 for DM, 36–121 cm for PH and 18–53 g for TGW (Table 2). The DH, an important trait for early sowing (Figure 2) indicated that the vegetative phase at locations in the NWPZ was longer than CPZ locations for all wheat genotypes tested. The longest vegetative phase was observed at Karnal, while the shortest at Dharwad (Figure 2). The best performing lines displayed almost similar ranges for the heading date in both zones (NWPZ and CPZ) of India. However, the best lines from CZ showed a slightly earlier heading day compared to lines selected from NWPZ.

### 3.2. Genotyping of Germplasm for Ppd and Vrn Genes

The result of genotyping were obtained as follows:

For the *Ppd-D1* specific primers, there were 93.6% of the insensitive allele in the CIMMYT lines, and 87% in the country lines (Appendix A). In the cases of *Vrn-A1*, *Vrn-B1* and *Vrn-D1,* 2.8%, 11.1% and 1.1% of recessive alleles were discovered in the CIMMYT lines, and 35.4%, 34.9% and 38.4% in the country lines, respectively (Appendix A). The percentage of heterozygotes ranged from 0.58% to 4.77%. We studied the effect of known Ppd and Vrn genes on the heading days (Figure 3, Appendix A). The Ppd-D1a gene was very significantly associated with heading in Dharwar (*p*-value = 3.63 × 10^−43^), DWR Karnal (*p*-value = 2.20 × 10^−18^), Indore (*p*-value = 3.72 × 10^−19^), Ludhiana (*p*-value = 2.89 × 10^−22^), Obregon (*p*-value = 9.31 × 10^−36^) and Pune (*p*-value = 1.31 × 10^−42^). Similarly, the VRN-A1 gene was significantly associated with heading in DWR Karnal (*p*-value = 2.19 × 10^−9^), Ludhiana (*p*-value = 2.68 × 10^−6^), Dharwar (*p*-value = 8.53 × 10^−21^), Obregon (*p*-value = 2.66 × 10^−17^) and Pune (*p*-value = 5.36 × 10^−20^). The VRN-B1 gene was also associated with heading in Dharwar (*p*-value = 1.42 × 10^−3^). We also observed clear differences in the days to heading means of the lines with sensitive and insensitive alleles at the Ppd-D1a gene and dominant and recessive alleles at the VRN-A1 and VRN-B1 genes.

In the case of *Vrn-B3*, of 2751 CIMMYT lines, 2730 lines showed recessive alleles. Likewise, in 461 country lines, 450 were recessive (Appendix A). There were no heterozygotes in CIMMYT lines, while four country lines showed this feature.

For Ppd-A1a, a total of nine insensitive genotypes and six heterozygous genotypes were detected in the CIMMYT lines (Appendix A).

Genotyping of GWM4167 (=WMS4167) associated with Ppd-B1 resulted in eight alleles for the CIMMYT lines (Appendix A) and seven alleles for the country lines (Appendix A). The flowering-delaying allele GWM4167-217bp was present in 589 CIMMYT lines and 51 country lines.

The genotyping of the Vrn-A2 linked GWM291 (=WMS291) resulted in eight alleles for the CIMMYT lines (Appendix A) and in 17 alleles for the country lines (Appendix A). The beneficial allele GWM291-176bp was present in 14 lines of the CIMMYT population and four lines of the country lines.

For the insensitivity locus *Ppd-B1* on chromosome 2BS, the insensitive genotype was discovered in 1206 (43.8%) of the CIMMYT lines and in 179 (38.8%) of the country lines (Appendix A).

### 3.3. Performance of Elite Lines in the Early (October) Sown Conditions

Based on selection and testing of best lines from the original set of 3322 diverse genotypes, the results obtained are given below.

The details summary of allele types, numbers and frequency based on DNA analysis of markers associated with Vrn and Ppd genes is given in Appendix A.

#### 3.3.1. Evaluation during 2013–2014 Crop Season—NWPZ

In the NWPZ zone, significant variation for yield and yield traits was observed among the 50 lines tested in the October sown trial at the three locations (Ludhiana, Karnal and New Delhi) (Table 3). This variation was also supported by the box plot for grain yield and yield traits (Figure 4). The biplot and dendrogram drawn for the three locations of NWPZ showed that the three locations showed different behavior for grain yield and plant height, but TGW and DM expressed high similarity in Ludhiana and Karnal (Appendix A).

The biplot for grain yield and yield traits showed that under October sowing of 2013–2014 in three locations of NWPZ, the most stable performance for grain yield was shown by genotypes: BMZ-NW-9, BMZ-NW-4 and the check variety BMZ-NW-7 (DPW 621-50) (Appendix A). However, the numerical value of mean grain yield showed that the best genotype was BMZ-NW-16, which was significantly superior to the check DPW 621-50, which performed better than the other check HD 2967, the most dominant variety of NWPZ covering more than half of the wheat area under this zone (Table 4). Although no other genotype was significantly superior to DPW 621-50, ten genotypes tested under October sowing in NWPZ were found significantly superior to HD 2967 (Table 4).

#### 3.3.2. Evaluation during 2013–2014 Crop Season—CPZ

Like the NWPZ, CPZ zone also showed significant variation for yield and yield traits among the 50 lines tested under early (October) sown conditions at the three locations (Indore, Pune and Dharwad) in 2013–2014 crop season (Table 5). The biplot and dendrogram drawn for the three locations of CPZ showed that the three locations showed different behavior for all four traits, although there was some similarity with Pune and Dharwad except for plant height (Appendix A).

The biplot for grain yield and yield traits showed that under October sowing of 2013–2014 in three locations of CPZ, the most stable performance for grain yield was shown by genotypes: BMZ-CPZ-12, BMZ-CPZ-4 and BMZ-CPZ-36 (Appendix A). However, the numerical value of mean grain yield showed that the best genotype was BMZ-CPZ-07 which was significantly superior to the check MACS 6222 which performed much better than the other check GW 322, the most dominant variety of CPZ (Table 6). Although no other genotype was significantly superior to MACS 6222, ten genotypes tested under October sowing in CPZ were significantly superior to GW 322 (Table 6).

Overall, several stable lines with grain yield advantage of more than 1.0 t/ha over the most popular varieties (HD 2967 in NWPZ and GW 322 in CPZ) were obtained in both zones of India (Table 4 and Table 6). This was an interesting evidence to demonstrate that there is ample possibility of obtaining wheat genotypes for October sowing and that in future varieties for early sowing, which are supposed to possess early heat tolerance, can be developed and released for farmers.

#### 3.3.3. Performance of Genotypes in 2014–2015 Crop Season in NWPZ

Like the 2013–2014 season, a significant variation was again noted for yield and yield traits in the 2014–2015 season among the 30 lines tested in the October sown trial at the three locations (Ludhiana, Karnal and New Delhi) (Table 3). The biplot and dendrogram drawn for the three locations of NWPZ showed that the three locations showed different behavior for grain yield and plant height, but TGW and DM expressed quite similarly in Ludhiana and Karnal (Appendix A). The PCA between traits within location and suggests strong interaction of environment on heading days (Appendix A) 

The impressive performance of some of the new lines in the NWPZ zone under October sowing was again demonstrated in the crop season 2014–2015 (Table 7). The average performance for grain yield and yield traits showed that under October sowing of 2014–2015 in three locations of NWPZ, the most stable performance for grain yield was shown by genotypes: BMZ-NW-16, BMZ-NW-7, BMZ-NW-9, BMZ-NW-4 and BMZ-NW-42 (Table 7). All these lines were significantly superior to the check variety HD 2967, which has been most popular among farmers in the years of these experimentations (Table 7). Their superiority in numerical terms over HD 2967 was in the range of 11–27%. Since performance of genotypes varied across locations as indicated by significant Loc*Geno interaction, genotypes that performed significantly superior to checks differed in each of the three locations.

## 4. Discussion

Since green revolution, the wheat breeding programs in different countries of South Asia have released a significant number of well adapted wheat varieties and thus have been able to sustain wheat production matching with the need of the regions. The collaboration between CIMMYT and South Asian countries have played a major role in this developing varieties adapted to variable conditions in different agro-ecological zones [1,29]. However, these varieties were released for normal (November) or late (December) sowings under different management conditions. In parts of Punjab, farmers used to plant mostly in the first fortnight of November, but a few as early as in the end of October. However, wheat planting much earlier, i.e., in the third week of October due to its various advantages under changing weather patterns continues to be a dream for the farmers. As the monsoon period is the major rainfall period, farmers desired to plant wheat earlier in the season in many parts of northwestern and central India to take advantage of the residual moisture and escape terminal heat stress in the month of March since if planted early, wheat would be past physiological maturity by that time [30]. However, whenever this was attempted, the wheat crop used developed too quickly under the warm/hot early seedling conditions, resulting in reduction in biomass, ear and grain numbers, with resultant decreases in yield. The reason being the prevalent wheat varieties being poorly adapted to warmer temperatures at juvenile stages. In fact, breeding was never done for heat tolerance at the juvenile stage, and hence there was no knowledge on this subject.

The results of this study showed that wheat lines with early heat tolerance can be obtained if diverse breeding lines are exposed to selection under October sowing [30]. The performance of top lines reached to almost 8 t/ha, which is remarkable compared to the performance of Elite Spring Wheat Yield Trial (ESWYT) lines under November sowing, which hardly exceeded 6 t/ha [31]. This performance was observed under similar fertilizer and irrigation management. Hence, the research results are expected to benefit wheat growers and consumers through this new type of wheat germplasm, which can perform well under early sowing. The sustainability of wheat production will be further enhanced by reduced pumping of water. Since there are machines available like Happy Seeder, super seeder and super straw management system in the combines, early planting is possible immediately after harvest [32]. This throws another advantage of reducing straw burning and the consequent emissions of greenhouse gases. Since varieties suitable for early sowing were not available in the past, there was no scope of immediate planting after rice harvest which mostly (about 90%) happens in the month of October. The availability of wheat varieties with early heat tolerance, therefore, is expected to generate all round benefits to the farmers and the sustainability of environment.

The superior performing genotypes under early sowing can be used in crossing program for further strengthening breeding for early heat tolerance in a systematic manner. Hence, the results of this study showing possibility of breeding for early heat tolerance will be applicable to many wheat growing areas worldwide. The ultimate users will be smallholder wheat farmers in both irrigated and rainfed situations in South Asia estimated to exceed 10 million wheat farming households, and through spin-off effects on technology development in other parts of the world [33]. Farmers will have options for more sustainable and productive farming systems, even under conditions of climate change. Intermediate users of the technology will include the All India Co-ordinated Wheat Improvement Project (AICWIP) through national wheat breeding program; and National Agriculture Research System (NARS) breeding programs in development of superior wheat varieties. Most NARS in the developing world have well-established linkages with CIMMYT and receive wheat nurseries and trials annually [20]; CIMMYT spring and winter bread wheat breeding programs that are partners in this project for development of lines for international distribution; and breeding programs in developed countries for use as parents in variety development [34]. This is likely at the organizations involved in this experimentation, but also others, as information and germplasm will be freely available.

The most common vernalization alleles in the project materials were the dominant spring alleles *Vrn-D1a*, followed by *Vrn-B1a*. The Japanese cultivar ‘Akakomugi’ is thought to be the donor parent of the *Vrn-D1a* allele in CIMMYT wheat [35], which was later transferred into early Green Revolution cultivars like ‘Lerma Rojo’ and ‘Sonora 64.’ These two cultivars are also thought to be the potential source of the *Vrn-D1a* allele in South and Southeast Asian wheat [35,36]. Furthermore, it has been concluded that the highest yield was predicted for varieties containing *Vrn-D1a* [37]. While the frequency of both alleles was above 90% in CIMMYT lines the frequency was lower in the set of global cultivars and also in a set of lines released in India [38].

Plant height is one of the crucial traits to understand cultivars superiority under early sown condition. However, we observed a poor correlation (0.109, *p* < 0.001; Appendix A) between plant height and grain yield, indicating that plant height did not play a significant role to get higher yield under early sown condition. However, it has been shown that early planted wheat plants gain slightly more height compared to those normally sown [39]. The dominant spring wheat allele *Vrn-A1a* was almost absent in CIMMYT wheat, but was present in 39% of global cultivar set. When released cultivars in India were evaluated for the *Vrn-1* genes, a high frequency (>60%) for the *Vrn-A1a* allele was observed [40]. No or very little variation in all datasets was observed for *Vrn-A2* and *Vrn-B3*. A slightly higher frequency of the *Vrn-A2* was selected for CZ.

The allele *Ppd-D1a* was the predominant photoperiod insensitive allele in the lines evaluated. The *Ppd-D1a* allele was introduced into CIMMYT wheat since Norman Borlaug shuttled germplasm between two contrasting environments in Mexico and exposed wheat materials to diverse photoperiods and temperatures. The *Ppd-B1* alleles showed the highest variation across datasets of genotypes investigated. A lower frequency of the insensitive allele *Ppd-B1a* was selected for NWPZ and CZ. At the same time, a higher frequency of the *Ppd-B1(217bp)* allele was selected for both NWPZ and CZ. In the set of global wheat cultivars, the frequency of the *Ppd-B1a* allele was similar than in the NWPZ and CZ selections. The frequency of the allele *Ppd-B1(217bp)* was lowest in the global cultivars set. The *Ppd-A1a* alleles were first described in durum wheat and are present CIMMYT bread wheat germplasm due to introgressions from synthetic hexaploid wheat, but with low frequency. Synthetic hexaploid is developed by creating a cross between *Aegilops tauschii* and durum wheat. Only 0.3% of all CIMMYT lines contained this allele, and none of those lines was further selected. The allele was not observed in the set of global cultivars.

It is a well noted history that since the early days of Green Revolution, the Indo-Gangetic Plains of South Asia has transformed this region from a food deficit to a food surplus region. At present this region produces about 15% of global wheat production and is inhabited by one sixth of the world population. Enough concerns have been expressed about this region being under threat from climate change, mainly heat and water stress, and may get converted to a sub-optimal wheat belt by the year 2050 [3]. A number of challenges in wheat production have been described for India [1] and whole of South Asia [2].

There are indications that useful variation for heat tolerance is available in the wheat gene pool [1,41,42]. The best way to breed for terminal heat tolerance in wheat has been to delay planting so that wheat populations are exposed to high temperatures from heading onwards and thereby there is high chance of selection for only those plants that have high terminal heat tolerance. Likewise, as shown in this study, early sowing can be done to select for lines that have tolerance to early high temperature. However, the reason for superior performance by a good number of lines under early sowing is not fully understood. There is possibility that it is combination of mild vernalization, superior agronomic performance and some unknown genes that might be supporting proper growth and tillering under early heat. In fact, the genetic control of adaptation response and heat tolerance is poorly understood. Major genes underlying adaptation have been identified in recent years, namely those controlling vernalization response, photoperiod sensitivity and development rate “earliness per se” [9,10,11,12]. To bridge the gaps in the knowledge of the effects of possible allele combination on adaptation in specific environments was attempted to address (Figure 3 and Appendix A). A linear model was fitted with days to heading in different environments as the y-variable and the alleles at the Ppd-D1a, VRN-A1, VRN-B1 and VRN-D1 genes as the x-variables. The effects of the alleles on heading in different environments and the p-values for the test of the significance of the effects were obtained using a two-tailed t-test. The *Vrn* genes determine the control of the spring/winter wheat difference by defining the chilling hours required by the wheat plant to be able to flower, while the *Ppd* genes play an important part in delaying flowering time in the spring after vernalization requirement has been satisfied [43]. The effects of the *Eps* loci may facilitate more subtle manipulation of the life cycle for regional adaptation. QTL studies have described various regions of potential *Eps* genes [13,14,44,45]. Allelic variation for some QTL, for example effects on 4B, 6A, 7D, appear to occur frequently in diverse germplasm. *Vrn*, *Ppd* and *Eps* genes additionally function in epistatic interaction [15]. Large numbers of allele combination will therefore be involved in determining the regulation of growth habit and optimal adaptation to a certain environment [46,47]. Further work will be required to understand the genetic basis of superior performance of wheat lines under early sowing.

Overall, the study resulted in proving the fact that it is possible to obtain wheat lines that can demonstrate significantly superior grain yield and agronomic traits under early sowing conditions of south Asia that has potential to give additional 1 t/ha yield under same agronomic management. Based on the benefits of these trials, Indian Council of Agricultural Research (ICAR), New Delhi initiated coordinated trials for early sowing in India for the first time with the purpose of breeding and encouraging wheat planting in the third week of October. Consequently, in the year 2020, three wheat varieties (DBW187, DBW303 and WH1270) were identified for release for early sown irrigated conditions in the NWPZ of India, the first case of this kind in India (https://www.aicrpwheatbarley.org/wp-content/uploads/2020/08/VIC-proceedings-2020.pdf, accessed on 15 September 2021).

## Figures and Tables

**Figure 1 genes-12-01808-f001:**
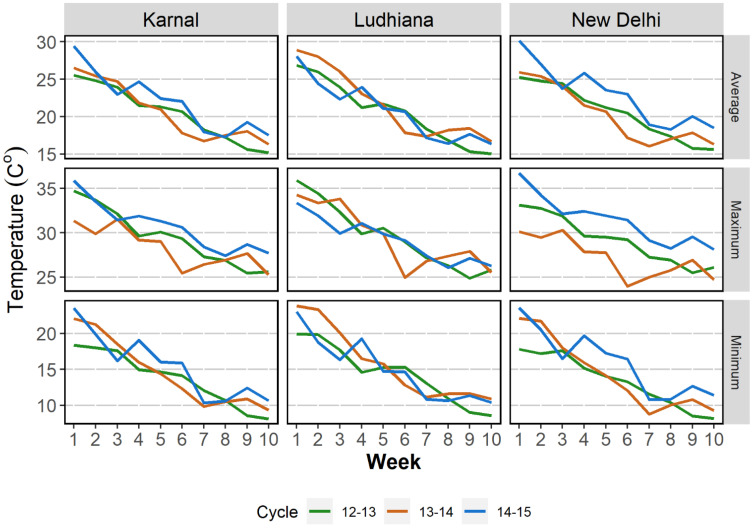
Weather parameters in the three (2012–2015) years of testing in northwest India.

**Figure 2 genes-12-01808-f002:**
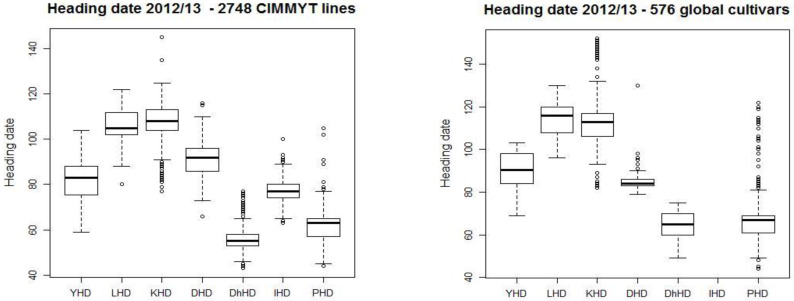
Box-plot for heading date of 3222 CIMMYT lines and global cultivars in Mexico and six different environments in India in 2012–2013. Y: Ciudad Obregon, L: Ludhiana, India, K: Karnal. D: Delhi, Dh: Dhawar. I: Indore, P: Pune. The standard error bars are shown as dotted lines and circles as outliers. The horizontal bar in the box is the median or 50^th^ percentile.

**Figure 3 genes-12-01808-f003:**
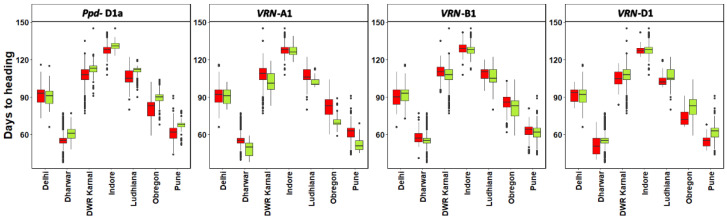
Effect of known Ppd and Vrn genes on the heading days. The horizontal bar in the box is the median or 50^th^ percentile. The recessive allele is denoted by zero while dominant is denoted by 1.

**Figure 4 genes-12-01808-f004:**
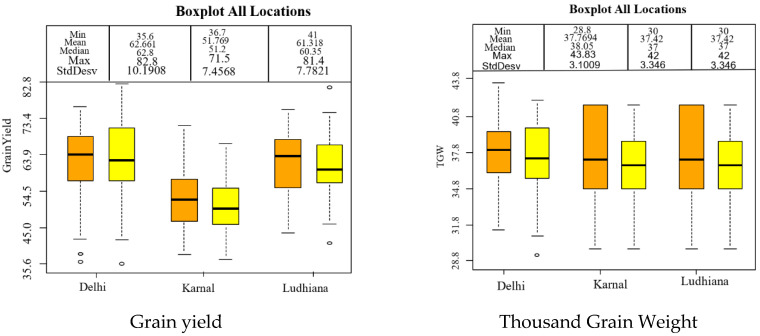
Box plot for grain yield and yield traits in early sown trial of 50 lines tested in three locations (Ludhiana, Karnal and New Delhi) of NWPZ of India in 2013–2014. The standard error bars are shown as dotted lines and circles as outliers. The horizontal bar in the box is the median or 50^th^ percentile.

**Table 1 genes-12-01808-t001:** Analysis of variance of five traits in 3226 genotypes of wheat when evaluated under six locations of India under early (October) sowing conditions.

Source	df	Mean Sum of Squares				
		GY	Heading	Maturity	Pl Height	TGW
Genotype	3326	173,725 **	2661.8 **	2574.6 **	1125.1 **	1339.0 **
Error	15,713	31,919	82.8	415.3	77.9	41.4
F value		5.44	32.14	6.20	14.44	32.33

** Significant at *p* < 0.0001.

**Table 2 genes-12-01808-t002:** Mean and variance for the five traits for 3226 genotypes of wheat when evaluated under six locations of India under early (October) sowing conditions.

	GY per Plot (t/ha)	Heading (Days)	Maturity (Days)	Pl Height (cm)	TGW (g)
Range	1.1–8.2	53–112	70–143	36–121	18–53
Var_Genotype	1.10	12.19	2.54	22.17	6.69
Var_Resid	324.9586	82.82	415.33	77.94	41.42
Mean	4.4651	83.72	124.37	90.00	43.83
LSD	1.4	7	16	7	5
CV	13.9	10.9	6.4	9.8	14.7
Heritability	0.200	0.4689	0.354	0.6306	0.4922

**Table 3 genes-12-01808-t003:** Analysis of variance of grain yield and yield traits of wheat genotypes when evaluated under three locations of NWPZ of India under early (October) sowing conditions in 2013–2014 and 2014–2015.

Source	df	Mean Sum of Squares (50 Genotypes 2013–2014)			
		GY	Heading	Pl Height	TGW
Loc	2	3527.04 **	7650.75 **	1614.72 **	4.06
Rep (Loc)	3	62.04	2.00	3.96	9.83
Genotype	49	172.66 **	28.18 **	63.42 **	15.33 **
Loc × Genotype	98	111.76 **	8.44 **	31.08 **	15.29 **
		**Mean Sum of Squares (30 Genotypes 2014–2015)**			
Loc	2	6591.66 **	9669.93	7957.21 **	988.24 **
Rep (Loc)	3	77.65	4.12	8.78	4.13
Genotype	30	132.95 **	10.94	3.77	9.20 **
Loc × Genotype		74.14 **	13.29	33.20 **	16.72 **

** Significant at *p* < 0.0001.

**Table 4 genes-12-01808-t004:** Performance of genotypes for grain yield under early sowing in 2013–2014 in three locations of the North West Plains Zone of India.

No.	Delhi			Karnal			Ludhiana			Mean of Locations		
	Line	GY t/ha	% Gain over LC	Line	GY t/ha	% Gain over LC	Line	GY t/ha	% Gain over LC	Line	GY t/ha	% Gain over LC
1	BMZ-NW-04	7.78	10.4	BMZ-NW-07	6.91	44.6	BMZ-NW-16	7.85	52.1	BMZ-NW-16	7.35	29.8
2	BMZ-NW-30	7.57	7.4	BMZ-NW-16	6.71	40.4	BMZ-NW-26	7.25	40.5	BMZ-NW-07	7.15	26.3
3	BMZ-NW-07	7.51	6.5	BMZ-NW-40	6.57	37.4	BMZ-NW-35	7.16	38.8	BMZ-NW-09	6.60	16.5
4	BMZ-NW-16	7.49	6.2	BMZ-NW-27	6.22	30.1	BMZ-NW-28	7.15	38.6	BMZ-NW-04	6.50	14.8
5	BMZ-NW-17	7.39	4.8	BMZ-NW-06	6.21	29.9	BMZ-NW-25	7.10	37.6	BMZ-NW-42	6.46	14.1
6	BMZ-NW-43	7.38	4.7	BMZ-NW-41	5.93	24.1	BMZ-NW-17	7.03	36.2	BMZ-NW-29	6.33	11.8
7	BMZ-NW-09	7.36	4.4	BMZ-NW-29	5.82	21.8	BMZ-NW-09	7.02	36.0	BMZ-NW-28	6.32	11.6
8	BMZ-NW-31	7.30	3.5	BMZ-NW-03	5.78	20.9	BMZ-NW-45	6.81	32.0	BMZ-NW-17	6.30	11.2
9	BMZ-NW-42	7.25	2.8	BMZ-NW-20	5.72	19.7	BMZ-NW-22	6.79	31.6	BMZ-NW-30	6.27	10.7
10	BMZ-NW-29	7.11	0.9	BMZ-NW-48	5.65	18.2	BMZ-NW-48	6.78	31.4	BMZ-NW-06	6.26	10.5
11	DPW 621-50 (C)	7.50		DPW 621-50 (C)	6.35		DPW 621-50 (C)	6.97		DPW 621-50 (C)	6.94	
12	HD 2967(C)	7.05		HD 2967(C)	4.78		HD 2967(C)	5.16		HD 2967(C)	5.66	
	LSD 5%	3.1			4.2			4.4			4.3	

Note: LC = Local Check (HD2967) is the most dominant variety of the NWPZ of India.

**Table 5 genes-12-01808-t005:** Analysis of variance of grain yield and yield traits of wheat genotypes evaluated at three locations of CPZ of India under early (October) sowing conditions in 2013–2014.

Source	df	Mean Sum of Squares (50 Genotypes 2013–2014)			
		GY	Heading	Pl Height	TGW
Loc	2	4890.66 **	17,617.04 **	8631.54 **	7482.93 **
Rep(Loc)	3	136.25	236.11 **	60.85	12.40
Geno	49	5025.41 **	3660.75 **	4553.41 **	3364.82 **
Loc × Geno	98	6924.13 **	1464.96 **	1694.79 **	992.25 **

** Significant at *p* < 0.0001.

**Table 6 genes-12-01808-t006:** Performance of genotypes for grain yield under early sowing in 2013–2014 in three locations of the Central and Peninsular Plains Zone of India.

No.	Dharwad			Indore			Pune			Mean of Locations		
	Line	GY t/ha	% Gain over GW 322	Line	GY t/ha	% Gain over GW 322	Line	GY t/ha	% Gain over GW 322	Line	GY t/ha	% Gain over GW 322
1	BMZ-CPZ-04	7.78 *	9.1	BMZ-CPZ-07	6.91 *	44.6	BMZ-CPZ-16	7.85 *	52.3	BMZ-CPZ-07	7.00 *	23.1
2	BMZ-CPZ-30	7.57 *	6.1	BMZ-CPZ-16	6.71 *	40.5	BMZ-CPZ-26	7.25	40.5	BMZ-CPZ-16	6.85	20.4
3	BMZ-CPZ-07	7.51	5.4	BMZ-CPZ-40	6.57	37.5	BMZ-CPZ-35	7.16	38.8	BMZ-CPZ-09	6.60	16.0
4	BMZ-CPZ-17	7.39	3.6	BMZ-CPZ-27	6.22	30.2	BMZ-CPZ-28	7.15	38.6	BMZ-CPZ-04	6.50	14.2
5	BMZ-CPZ-43	7.38	3.4	BMZ-CPZ-06	6.21	30.0	BMZ-CPZ-25	7.10	37.8	BMZ-CPZ-42	6.46	13.5
6	BMZ-CPZ-09	7.36	3.2	BMZ-CPZ-41	5.93	24.1	BMZ-CPZ-17	7.03	36.4	BMZ-CPZ-29	6.33	11.2
7	BMZ-CPZ-31	7.30	2.4	BMZ-CPZ-29	5.82	21.9	BMZ-CPZ-09	7.02	36.2	BMZ-CPZ-28	6.32	11.2
8	BMZ-CPZ-42	7.25	1.7	BMZ-CPZ-03	5.78	21.0	BMZ-CPZ-45	6.81	32.0	BMZ-CPZ-17	6.30	10.8
9	BMZ-CPZ-29	7.11	−0.2	BMZ-CPZ-20	5.72	19.7	BMZ-CPZ-22	6.79	31.7	BMZ-CPZ-30	6.27	10.2
10	BMZ-CPZ-11	7.03	−1.4	BMZ-CPZ-48	5.65	18.2	BMZ-CPZ-48	6.78	31.5	BMZ-CPZ-06	6.26	10.1
11	MACS6222 (C)	7.20		MACS6222 (C)	6.35		MACS6222 (C)	6.97		MACS6222 (C)	6.84	
12	GW 322 (C)	7.13		GW 322 (C)	4.78		GW 322 (C)	5.16		GW 322 (C)	5.69	
	LSD 5%	3.7			3.4			4.3			3.3	

Note: The check GW 322 is the most dominant variety of the CPZ of India; * significantly superior over both checks.

**Table 7 genes-12-01808-t007:** Performance of genotypes for grain yield under early sowing in 2014–2015 in three locations of the North West Plains Zone of India.

No.	Delhi			Karnal			Ludhiana			Mean of Locations		
	Line	GY t/ha	% Gain over HD 2967	Line	GY t/ha	% Gain over HD 2967	Line	GY t/ha	% Gain over HD 2967	Line	GY t/ha	% Gain over HD 2967
1	BMZ-NW-2015-6	8.00	22.0	BMZ-NW-2015-20	7.74	25.5	BMZ-NW-2015-25	7.92	34.7	BMZ-NW-16	7.89	27.2
2	BMZ-NW-2015-20	7.56	15.3	BMZ-NW-2015-10	7.36	19.4	BMZ-NW-2015-21	7.55	28.3	BMZ-NW-07	7.49	20.8
3	BMZ-NW-2015-17	7.40	12.9	BMZ-NW-2015-23	7.21	16.8	BMZ-NW-2015-26	7.49	27.5	BMZ-NW-09	7.37	18.8
4	BMZ-NW-2015-9	6.99	6.6	BMZ-NW-2015-28	7.03	13.9	BMZ-NW-2015-28	7.23	22.9	BMZ-NW-04	7.08	14.2
5	BMZ-NW-2015-28	6.73	2.7	BMZ-NW-2015-18	6.76	9.6	BMZ-NW-2015-27	7.20	22.5	BMZ-NW-42	6.90	11.2
6	BMZ-NW-2015-24	6.60	0.6	BMZ-NW-2015-25	6.73	9.0	BMZ-NW-2015-12	7.09	20.5	BMZ-NW-29	6.80	9.7
7	BMZ-NW-2015-13	6.42	−2.2	BMZ-NW-2015-24	6.65	7.7	BMZ-NW-2015-5	6.87	16.8	BMZ-NW-28	6.64	7.1
8	BMZ-NW-2015-25	6.39	−2.6	BMZ-NW-2015-22	6.55	6.2	BMZ-NW-2015-18	6.80	15.7	BMZ-NW-17	6.58	6.1
9	BMZ-NW-2015-10	6.29	−4.1	BMZ-NW-2015-16	6.23	0.9	BMZ-NW-2015-3	6.71	14.1	BMZ-NW-30	6.41	3.3
10	BMZ-NW-2015-18	6.23	−5.0	BMZ-NW-2015-6	6.15	−0.3	BMZ-NW-2015-14	6.60	12.2	BMZ-NW-06	6.33	2.0
11	KACHU #1 (C)	6.79		KACHU #1 (C)	5.78		KACHU #1 (C)	6.08		KACHU #1 (C)	6.22	
12	HD 2967 (C)	5.56		HD 2967 (C)	5.17		HD 2967 (C)	4.88		HD 2967 (C)	5.20	
	LSD 5%	4.5			5.3			3.5			4.1	

Note: The check HD2967 is the most dominant variety of the NWPZ of India.

## Data Availability

The data will be made available on request. Kindy contact the corresponding author.

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
