# Peer review of "Juvenile Heat Tolerance in Wheat for Attaining Higher Grain Yield by Shifting to Early Sowing in October in South Asia"

_genes, 2021, doi:10.3390/genes12111808_

Round 1

Reviewer 1 Report

The manuscript “Juvenile heat tolerance – search for a new trait in wheat for attaining higher yields by shifting to earlier sowing in October in South Asia” presents study of big wheat collection performance under earlier sowing in South Asia. Also, authors presented genotyping of this collection for major flowering genes.

In general presented study has value for readers, but required major revision:

  1. I suggest to change title of manuscript. In my understanding “trait” is calculated or directly measured phenotypic observation. Authors did not calculate any index or “new trait” in presented work.
  2. Main loss of manuscript is absence of data for tested collection sown under normal dates. I know that it is big work, but it is difficult to judge performance of identified superior lines without comparison to sowing at normal dates. I recommend authors to account such pitfall in future and to use “control” conditions at least for selected genotypes.
  3. Big work was done for genotyping of major flowering gene in studied collection. But authors did not presented any analysis that connected obtained genotyping with heading dates. I would like to see such analysis that can help to identify favorable alleles of studied genes for earlier sowing in South Asia. Additionally, it will be great to add excel table with names of wheat lines and alleles of studied genes in supplementary material.
  4. Authors mention “Green revolution” in text, but did not provide information about major gene of Green revolution – Rht 1. I believe that plant height is critical trait for heat tolerance as well as sensitivity to gibberellin. I would like to see such information, because it is crucial to understanding what cultivars are superior under studied conditions, dwarf or tall. In case of absence information about RHT-1, I suggest to make analysis for at least available phenotypic data of plant height.
  5. I would like to see more advance statistical analysis of relationships between studied traits such as correlation analysis and PCA.
  6. There are couple well known and widely used indexes of trait stability are available for analysis. I suggest to authors to use one or few of them in addition to presented PCA analysis.

Minor changes:

  1. There are problems in Figure 3 as some black lines. I recommend to present clearer figure.
  2. I suggest to move dendrogram and biplot figures to SM.
  3. I would like to see CI units (for example ton per hectare) in presented tables. It is “q/ha” now.
  4. There are multiple terminology used in tables presented analysis of variance. Entry, geno and genotype. I recommend to use only one, for example “genotype”.
  5. There is typo in title of figure S4.”CIMMYT”, but I expected “VRN-D1”.
  6. Supplementary figures are ungrouped parts of graphs and tables. I suggest to grouped them and add as one picture per figure.

Reviewer 2 Report

The study has been conducted well. However, following correction are needed:

Line 262: In Table 2. the mean value for GY is out of the Range. Correct whichever is wrong.

Line 305: Figure 3. Mention the units of the traits as well. Also do not understand the point of these slanted lines either explain or remove them. Figures should be clean.

Line 326: "Dharwad" is written as "Dharwar" in the subsequent figures, use the correct and single uniform term in the whole manuscript.

Line 425: did not find the full form of the abbreviation NAR in the text, mention in the first instance it is used. 

Author Response

The study has been conducted well. However, following correction are needed:

Line 262: In Table 2. the mean value for GY is out of the Range. Correct whichever is wrong.

Correction done. There was typing mistake of decimal.

Line 305: Figure 3. Mention the units of the traits as well. Also do not understand the point of these slanted lines either explain or remove them. Figures should be clean.

Done. Explained in the figure.

Line 326: "Dharwad" is written as "Dharwar" in the subsequent figures, use the correct and single uniform term in the whole manuscript.

Correction done

Line 425: did not find the full form of the abbreviation NAR in the text, mention in the first instance it is used. 

This “National Agricultural Research”. Correction done

Reviewer 3 Report

The manuscript "Juvenile heat tolerance – search for a new trait in wheat for attaining higher yields by shifting to earlier sowing in October in South Asia" is fitting well with the scope of the Journal and of the special issue.

The subject matter is very interesting and very important for its implications related to climate changes and food security. The experimental design allow to obtain interesting results for future breeding and selection.

The manuscript needs some improvements, detailed below:

  1. Please check carefully the text for minor English revisions, such as ")","(" (lines 44 and 55),".." at line 503 and "of by" at line 441.
  2. Sometimes you use the paper number as subject of the sentence, that is unpleasant, generally speaking, (examples at lines 55, 162, 443 and 438) and in other cases the author name (at line 166 and 441). Please try to harmonize.
  3. Furthermore [38] corresponds to Singh et al., 2014, while [39] corresponds to Singh et al., 2007 and, in any case, [38] needs to be cited before [39] (look at lines 441-443), please make the right corrections.
  4. Figure 9 is missing in the .pdf file, but is present in figures.docx file.
  5. The URL at line 502-503 is incorrect due to the bar added for line division.

Author Response

The subject matter is very interesting and very important for its implications related to climate changes and food security. The experimental design allow to obtain interesting results for future breeding and selection.

The manuscript needs some improvements, detailed below:

  1. Please check carefully the text for minor English revisions, such as ")","(" (lines 44 and 55),".." at line 503 and "of by" at line 441.

All corrections done.

  1. Sometimes you use the paper number as subject of the sentence, that is unpleasant, generally speaking, (examples at lines 55, 162, 443 and 438) and in other cases the author name (at line 166 and 441). Please try to harmonize.

Correction done

  1. Furthermore [38] corresponds to Singh et al., 2014, while [39] corresponds to Singh et al., 2007 and, in any case, [38] needs to be cited before [39] (look at lines 441-443), please make the right corrections.

Correction done

  1. Figure 9 is missing in the .pdf file, but is present in figures.docx file.

Figure included

  1. The URL at line 502-503 is incorrect due to the bar added for line division.

The bar is added automatically by the text editor. After accepting all changes available in track change, this will be auto-corrected. However, if issue persists, we will provide URL as well as insert a link for online version.

Round 2

Reviewer 1 Report

The manuscript “Juvenile heat tolerance trait in wheat for attaining higher grain yield by shifting to early sowing in October in South Asia presents revised manuscript of big wheat collection performance under earlier sowing in South Asia. Also, authors presented genotyping of this collection for major flowering genes.

Authors made casually minor changes in revised manuscript and mostly ignored major comments and manuscript still has few serious pitfalls that should be corrected in my understanding:

  1. Manuscript has two separated parts without appropriated connection between them: genotyping of flowering genes and field evaluation of studied wheat collection. I want to cite sentence from manuscript “However, there are major gaps in the knowledge of the effects of possible allele combination on adaptation in specific environments”. Authors have phenotypic data for heading from multiple location and years for whole collection and selected superior lines, but manuscript does not have proper analysis of effects flowering genes allele on obtained phenotypes. For example authors mention in discussion  frequency of Ppd genes alleles in selected genotypes, but there is no information in results part. I strongly recommend put efforts to connect genotyping data and field evaluation or delete genotyping part from manuscript.
  2. Authors did not provide detailed information about studied collection. I think that it should be Excel table with all studied lines with their names (if possible), geographic origin, pedigree and type (cultivar, breeding line). Without such information presented results of allele distribution of flowering genes are non-useful. Such table can combine data of flowering genes alleles as well.
  3. Proper statistical analysis of relations between studied traits is missing in manuscript. It required detailed correlation analysis and PCA (between traits, not between locations) of studied traits for whole collection and selected genotypes. ANOVA tables showed strong effects of locations on studied traits that required separated analysis for locations.

Minor changes:

  1. Information about used statistical software should be added to M&M part.
  2. Multiple terminology used in tables still presents in manuscript, for example table 2 has “Var_Entry” and table 5 has “Geno”.
  3. Figure 3 has double boxplots without proper explanation.